# Trajectories of Adjustment Disorder and Well-Being in Austria and Croatia during 20 Months of the COVID-19 Pandemic

**DOI:** 10.3390/ijerph20196861

**Published:** 2023-09-29

**Authors:** Irina Zrnić Novaković, Alina Streicher, Dean Ajduković, Marina Ajduković, Jana Kiralj Lacković, Annett Lotzin, Brigitte Lueger-Schuster

**Affiliations:** 1Department of Clinical and Health Psychology, Faculty of Psychology, University of Vienna, 1010 Vienna, Austria; alina.streicher@univie.ac.at (A.S.); brigitte.lueger-schuster@univie.ac.at (B.L.-S.); 2Department of Psychology, Faculty of Humanities and Social Sciences, University of Zagreb, 10000 Zagreb, Croatia; dean.ajdukovic@ffzg.hr (D.A.); jana.kiralj@gmail.com (J.K.L.); 3Department of Social Work, Faculty of Law, University of Zagreb, 10000 Zagreb, Croatia; majdukov@pravo.hr; 4Department of Psychiatry and Psychotherapy, University Medical Center Hamburg-Eppendorf, 20246 Hamburg, Germany; a.lotzin@uke.de; 5Department of Psychology, MSH Medical School Hamburg, 20457 Hamburg, Germany

**Keywords:** adjustment disorder, well-being, COVID-19, pandemic, mental health, latent growth curve modelling, LGCM

## Abstract

The present study aimed to investigate the trajectories of adjustment disorder (AD) symptoms and well-being over 20 months of the COVID-19 pandemic in Austria and Croatia. Further objectives of this study were to examine whether sociodemographic characteristics and the symptoms of anxiety and depression could predict these trajectories. As part of the pan-European ESTSS ADJUST study, *N* = 1144 individuals were recruited using convenience sampling and assessed four times between June 2020 and January 2022 through an online survey. Latent growth curve modelling was applied to estimate the trajectories of AD symptoms and well-being. Over time, the prevalence of probable AD varied between 9.8% and 15.1%. The symptoms of AD tended to increase, whereas well-being tended to decrease. According to the majority of the models tested, women, participants from Austria and those with lower income had higher initial AD symptoms, whereas older participants and those from Croatia had higher initial well-being. In all models and at all timepoints, anxiety and depression significantly predicted AD and well-being scores. Overall, our study points to several predictors of AD and well-being and indicates high variability in people’s reactions to the pandemic. Psychosocial support for the general population is needed during pandemics and similar crises, with a special focus on vulnerable groups.

## 1. Introduction

The past three years have been marked by the COVID-19 pandemic and its repercussions. Compromised mental and physical health, disrupted routines and changed social dynamics, along with financial losses and work-related challenges, are just some of the stressors people have been repeatedly exposed to (for an overview of stressors related to the COVID-19 pandemic, see [1]). Due to its long-lasting and potentially traumatic impact [2], COVID-19 has been described as “continuous traumatic stress” [3]. According to the literature, adjustment disorder (AD) can best capture a maladaptive reaction to pandemic-related stress or trauma [4,5]. However, longitudinal studies on changes in AD symptoms over time are scarce, and potential cross-country differences amidst COVID-19 are still poorly understood. Acknowledging that AD symptoms are likely to develop in response to pandemics but also to other global challenges (e.g., the climate crisis, inflation, wars and refugee crises), the present study aims to better understand their trajectories and underlying factors in Austria and Croatia. Despite their geographical proximity, historical ties and cultural and economic exchanges [6], these two European Union countries differ in terms of their socioeconomic situations and degrees of life satisfaction [7,8], as well as their responses to the COVID-19 pandemic. For context, Croatia implemented stricter measures at the beginning of the pandemic, whereas government measures in Austria were stricter in the later stages of the pandemic [9]. 

### 1.1. Adjustment Disorder Amidst COVID-19

AD can be defined as a maladjustment to a stressful event or severe changes in life, such as job loss, financial strain or the illness of a loved one [10,11]. It is characterised by a preoccupation with the stressor and a failure to adapt which severely impacts one’s functioning. According to the World Health Organisation [12], the symptoms of AD usually resolve within six months. However, the symptoms might last longer if the stressor persists longer [13], which was the case for the COVID-19 pandemic and its consequences. 

During the pandemic, international studies researching different samples reported prevalence rates of AD ranging from 8.2% to 61.3% [14,15]. In older people, diagnoses of AD were more often documented before (2017–2019) than during the pandemic (2020 [16]). Gender differences were also found, with 40.7% of women compared to 31% of men scoring above cut-off for probable AD in a large sample of Georgian participants [17]. 

Apart from the above-mentioned sociodemographic characteristics, adjustment difficulties have been linked to COVID-related stressors, such as social restrictions, governmental pandemic management, work and financial difficulties [18]. Notably, the mental health impact of these stressors varied between countries, with countries having had more experience with stress and trauma showing better outcomes [19]. Different pandemic loads in terms of the stringency index and incidence and death rates could also explain cross-country differences in some mental health outcomes, e.g., anxiety, depression and post-traumatic stress disorders [20,21]. However, associations between pandemic load and AD symptoms have not yet been investigated. 

In general, some findings on factors associated with AD symptoms are inconsistent, whereas others are more robust. For instance, older participants were found to be at risk for developing AD in some studies [18], while a protective role of older age was proposed in other studies [22]. In contrast, a large body of research has shown that AD symptoms are associated with different emotional responses to the pandemic, including the fear of infection [23], COVID-19 concerns [17] and depression symptoms [24]. 

### 1.2. Trajectories of Adjustment Disorder Amidst COVID-19

To date, only a small number of studies have investigated changes in AD symptoms over the course of the COVID-19 pandemic. There is evidence that the symptoms of AD consolidated in autumn 2020 [25], although a German study comparing AD symptoms in the spring and autumn of 2020 found no significant differences [26]. 

A representative study conducted in Israel in the summer of 2020 (before the second lockdown) and in late autumn in 2020 (after the second lockdown) detected four trajectories of AD symptoms, suggesting different patterns of adjustment to the pandemic [27]. Remarkably, there is only limited knowledge of positive psychological adjustments to the pandemic over time since the indicators of positive mental health, such as well-being, have not yet been included in the research on AD amidst COVID-19. Thus far, scientific attention has mainly been directed toward vulnerable groups and risk factors for probable AD (e.g., [23,28]).

Shifting the focus toward indicators of positive mental health and the course of AD symptoms in the later phases of COVID-19 could yield a more differentiated picture of the psychological adjustment to the pandemic. Longitudinal studies over a longer period of time are particularly beneficial given that some individuals may experience adjustment difficulties only in the later phases of the pandemic [28]. Moreover, the changing context of the pandemic in terms of the strictness of COVID-19 restrictions and the incidence and death rates might foster an understanding of changes in both adjustment and well-being, yet these interrelations have not been explored. 

### 1.3. Current Study

In summary, the existing research on AD amidst COVID-19 has some limitations. Firstly, given that the majority of studies were conducted in the first months of the pandemic and covered intervals of several weeks or a few months, little is known about the trajectories of AD symptoms over a longer period and in later phases of the pandemic. Secondly, the co-development of well-being and AD symptoms during the pandemic is still poorly understood. The inclusion of sociodemographic characteristics (e.g., income and education) and mental health indicators (e.g., anxiety symptoms) might increase knowledge about this co-development [29,30,31]. Finally, a significant amount of uncertainty still exists with respect to cross-country differences, and a closer examination of the course of the pandemic in different countries might help to explain the development and severity of adjustment difficulties. 

To address these research gaps, the present study examined the trajectories of AD symptoms in Austria and Croatia over 20 months of the COVID-19 pandemic. Both countries are part of the pan-European ADJUST study on stressors, coping and the symptoms of AD [18,32]. In addition to the three waves of assessments conducted in all ADJUST member countries, a fourth wave was launched in Austria and Croatia in December 2021–January 2022. The aim was to assess psychological reactions to the pandemic one year after the large spike in COVID-19 cases and deaths over the winter months of 2020/2021 [33,34].

Several factors provide a basis for comparing Austrian and Croatian participants. Firstly, life satisfaction and quality of life have been higher in Austria than in Croatia in previous years, which could largely be explained by the higher gross domestic product (GDP) per capita among Austrians [35,36]. Secondly, the countries differ in terms of mental health, with Austrian citizens experiencing a higher mental health burden [37]. Thirdly, Austria and Croatia have been affected by COVID-19 to different extents, with, e.g., higher numbers of confirmed deaths being reported in Croatia in the second year of the pandemic [38]. Government policies for managing COVID-19 also differed between the countries [34]. For these reasons, a comparison of Croatian and Austrian data might shed light on the effects of a variety of socioeconomic and contextual factors on adjustment trajectories.

### 1.4. Aims and Research Questions

The present study set out to explore the trajectories of AD and well-being in the first two years of the COVID-19 pandemic using latent growth curve modelling. Our main aims were to answer the following research questions: How do the trajectories of AD symptoms and well-being develop over time in Austria and Croatia?To what extent can sociodemographic characteristics (country, age, gender, income and education) predict the trajectories of AD and well-being?To what extent can the symptoms of anxiety and depression predict the trajectories of AD and well-being?

Acknowledging that people are capable of coping with stressful situations and adapting to changing situational demands [39,40], we hypothesised that the symptoms of AD would decrease, whereas well-being would increase over time. Based on the existing evidence, we further hypothesised that sociodemographic characteristics and the symptoms of anxiety and depression would predict the trajectories of AD and well-being.

## 2. Materials and Methods

The present study is part of the international ADJUST COVID-19 study involving 11 European countries [32]. For further information, please see the study’s website: https://estss.org/adjust/ (accessed on 26 September 2023) and the corresponding OSF registry where the study was preregistered: https://doi.org/10.17605/OSF.IO/8XHYG (accessed on 26 September 2023). The data used for the secondary analysis described in this paper were collected from the general populations of Austria and Croatia at four measurement timepoints. As described in the introduction, only these two countries launched a fourth assessment wave in the course of the ADJUST study. Though they are closely related, e.g., in terms of cultural and economic exchanges [6], Austria and Croatia differ in several aspects [35,38], thus forming an interesting basis for comparison. The periods of data collection and information about the COVID-19 situation in both countries at each timepoint are presented in Table 1. 

### 2.1. Procedure and Participants

Over the course of the ADJUST study (i.e., the primary analysis), all assessment waves were conducted as online surveys using *LimeSurvey*, an online tool for creating anonymous surveys and questionnaires in line with the General Data Protection Regulation (GDPR; www.limesurvey.org, accessed on 31 August 2023). We used several strategies to recruit participants for the first assessment wave (T1). For instance, our study invitation was distributed among the staff of major companies and professional associations, as well as among members of numerous leisure and interest groups. We also shared the study link on social media and distributed flyers in public spaces. A detailed overview of the recruitment strategies can be found at www.ncbi.nlm.nih.gov/pmc/articles/PMC8725769/bin/ZEPT_A_1964197_SM8752.docx (see Supplement 2, accessed on 26 September 2023). The survey was accessible via a website link or a QR code. On the first page of the survey, potential participants could read important information about the study (the aims of the study, data management and the right to withdraw) and, if interested, provide their informed consent. 

To participate in the present study (i.e., the secondary analysis), the following criteria were required: (1) the participants needed to be at least 18 years old, (2) have good command of the German or Croatian language, (3) live in Austria or Croatia, (4) agree to participate in the study, and (5) respond to at least two assessment waves. In Austria, ethical approval for the study was granted by the Ethics Committee of the University of Vienna (date of approval: 29 May 2020, reference number: 00554), whereas the Ethics Committee of the Department of Psychology in the Faculty of Humanities and Social Sciences at the University of Zagreb granted ethical approval in Croatia (date of approval: 21 May 2020). Prior to the initiation of the survey, informed consent was obtained online from all participants.

At baseline (T1), *N* = 1001 valid cases were obtained in Austria and *N* = 1787 were obtained in Croatia, out of which 52.5% Austrian participants (*n* = 525) and 57.8% Croatian participants (*n* = 1032) agreed to take part in the following waves. These participants were re-contacted after approximately 6, 12 and 18 months and asked to participate in further study waves (T2, T3 or T4, respectively). 

In total, *N* = 1144 participants (*M_age_* = 44.0, *SD_age_* = 13.5), who were predominantly female (73.5%) met the inclusion criteria and were included in the present analysis. Table 2 shows important characteristics of the study participants.

### 2.2. Measures

An overview of the measures included in the ADJUST study can be found in the respective study protocol [32]. For the present analysis, the following measures were used in the German and Croatian language, respectively.

The 8-item Adjustment Disorder New Module (ADNM-8 [41]) screens for life stressors and AD symptoms according to the 11th version of the International Classification of Diseases (ICD-11 [12]). It consists of four items on preoccupation and four items on adjustment difficulties; each are scored on a 4-point Likert scale ranging from 1 (*never*) to 4 (*often*). Higher scores indicate more intense AD symptoms, and a score of >22 represents a cut-off for probable AD. The ADNM-8 has shown good psychometric properties [41,42] and has been used to assess AD symptoms during the pandemic [28,43]. 

The 5-item World Health Organization Well-Being Index (WHO-5 [44]) measures subjective well-being on a 6-point Likert scale ranging from 0 (*at no time*) to 5 (*all of the time*). The total raw score, with possible values between 0 and 25, is multiplied by 4, resulting in a final score between 0 and 100, with the latter representing the highest degree of well-being. A score below 51 suggests poor well-being and might be indicative of depressive symptomatology [45,46]. The high validity and applicability of the WHO-5 have been demonstrated across different countries and study samples [46], and the scale has been widely used in the pandemic context [20,47].

The 4-item Patient Health Questionnaire (PHQ-4 [48]) measures the core symptoms of depression and anxiety on a 4-point Likert scale (ranging from 0 = *not at all* to 4 = *more than half the days*). It comprises a 2-item depression scale (the 2-item Patient Health Questionnaire (PHQ-2)) and a 2-item anxiety scale (the 2-item Generalized Anxiety Disorder Scale (GAD-2)). A score of >2 on one of these scales is used as cut-off point for possible depression or anxiety. All three scales (i.e., the PHQ-4, PHQ-2 and GAD-2) have shown good reliability and validity [49] and have been applied in various COVID-19 studies [50].

In the present study, each scale had a high level of internal consistency across all measurement timepoints, ranging from 0.81 to 0.92 (for details, please see the Appendix A). Depending on the measurement, Cronbach’s α or the Spearman–Brown coefficient was used, as appropriate [51]. In addition to the described scales, self-constructed items were used to assess the sociodemographic characteristics of the participants. Gender, age, education and income were included in the main analysis as these variables were found to be predictive of AD and/or well-being in previous studies [18,52,53,54] and might explain the differences between mental health outcomes in Austria and Croatia. 

### 2.3. Data Analysis

The analyses were conducted using IBM SPSS Statistics (version 27 [55]) and RStudio (version 2022.12.0.353 [56]). After performing descriptive and missing data analyses (using the Mann–Whitney *U* test and the Welch *t*-test), measurement invariance was assessed. Cochran’s Q test, followed by a post hoc analysis using Dunn’s procedure, was conducted to explore whether the percentage of participants at risk for AD, depression and anxiety was different at the different timepoints. The complete dataset without missing values was used for this analysis (*n* = 531). In addition, we conducted a linear regression analysis to check whether the anxiety and depression scores attained at T1 could predict AD symptoms and well-being scores at T4. With this exploratory analysis, we wanted to indirectly account for the possible predictive value of pre-existing mental health symptoms since no reliable pre-pandemic data were available in the study.

Following these preparatory steps, latent growth curve models (LGCMs) of different levels of complexity were tested. General rules for evaluating model fit were used for all analyses [57,58].

#### 2.3.1. Measurement Invariance

Measurement invariance considers the psychometric equivalence of a construct over time or across different groups [59]. The assessment of invariance is an important step prior to testing group differences (e.g., cross-country differences, as in this study) or comparing different measurement occasions (e.g., changes over time, as in this study). Therefore, we assessed *invariance over populations* by comparing the Austrian and Croatian samples, and we assessed *longitudinal measurement invariance* by comparing the data collected at each of the four measurement timepoints. 

We tested measurement invariance in the structural equation modelling (SEM) framework using a confirmatory factor analysis (CFA [60]) for all relevant measures (i.e., the ADNM-8, WHO-5, PHQ-2 and GAD-2). Following the general guidelines regarding estimation methods [60,61], we used the weighted least square mean and variance adjusted (WLSMV) estimator for testing the invariance of measures with four or fewer answering options (i.e., the ADNM-8, PHQ-2 and GAD-2) and the robust maximum likelihood (MLR) as an estimator for testing the invariance of measures with more than four answering options (i.e., the WHO-5). To handle missing data, listwise deletion was used in combination with the WLSMV, and the Full Information Maximum Likelihood (FIML) in combination with the MLR [60]. Parcelling was applied for the analysis of the ADNM-8 as it was the only measure with more than five items.

Given that all measures used in the study are well-established, we assumed psychometric equivalence across groups and over time. Therefore, we applied the *model-building approach* (also known as the *constrained baseline approach*) to assess measurement invariance, using the most restricted model (i.e., strong invariance) as the baseline [60]. This analysis was performed using lavaan, an open source R package (version 0.6.12 [62]). 

#### 2.3.2. Latent Growth Modelling

Latent growth curve (LGC) modelling is a statistical method through which within-person changes (i.e., intra-individual patterns of change) and between-person differences (i.e., inter-individual variability) over time can be estimated [63,64,65]. Thus, LGC modelling allows one to examine individual differences in a longitudinal design (i.e., individual latent trajectories), which is not feasible using conventional methods, such as an ANOVA. Another advantage of LGC modelling is its ability to handle missing data effectively [66]. 

The essential elements of all LGCMs are *fixed* and *random effects*. Fixed effects are factors estimated from the data, namely the *mean intercept* (i.e., starting point) and the *mean slope* (i.e., the increment or rate of change). Jointly, these factors define the average latent trajectory for the entire sample. On the other hand, random effects capture the variance around the average trajectory, i.e., the differences in the starting point and the rate of change across individuals. Taken together, the fixed and random effects allow for the description of the general characteristics of the growth curve (i.e., trajectory) for the entire sample and the subgroups within this sample [63,64]. 

In the present study, LGC modelling was performed within the SEM framework [67], using lavaan (version 0.6.15 [62]). Given that a linear change was hypothesised and the four timepoints were approximately equally spaced, the following coding scheme was used to represent the slope: 0, 1, 2, 3. For the intercept, the factor loadings for four measurement timepoints were set to 1 (i.e., 1, 1, 1, 1). In line with the current recommendations [64,68], the MLR was the preferred estimator for all analyses. A state-of-the-art method of handling missing data, FIML, was used to deal with partially missing data [69]. 

A series of LGCMs were used to assess changes in AD symptoms (determined via the ADNM-8) and well-being (determined via the WHO-5) over time:**Univariate models**: Firstly, univariate unconditional models were estimated to explore changes over time in AD and well-being separately. Next, the *time-invariant covariates* (TICs [64]) age, gender, country, education and income were included as predictors of the variability in the intercept and slope in two conditional models for AD symptoms and well-being, respectively. In a third step, depression (determined via the PHQ-2) and anxiety (determined via the GAD-2) scores were added as *time-varying covariates* (TVCs [64]).**Multivariate models**: The same procedure was repeated for multivariate models (i.e., joint models for AD symptoms and well-being) to investigate how AD symptoms and well-being simultaneously unfold over time [65]. The final model included all TICs and TVCs to explore whether these can explain the residual variance in the joint model. Moreover, the covariance between the intercept and slope was estimated for each primary outcome. Residuals between the primary outcomes were allowed to correlate within one timepoint. Figure 1 shows the proposed multivariate model of AD symptoms and well-being with all predictors and covariances.**Multivariate models by country:** In the last step, the final model was executed in the Austrian and Croatian samples separately in order to explore possible differences in the growth trajectory and predictors between the two countries. For this final analysis, the predictor “country” was obsolete; thus, it was excluded from the analysis. The other model specifications remained unchanged.

## 3. Results

### 3.1. Participant Flow and Missing Data

Participants who did not meet the inclusion criteria and those who did not provide answers for the primary outcome measures were excluded from the analysis, resulting in a total of *N* = 1144 participants (*n_Austria_* = 415; *n_Croatia_* = 729). Detailed information about participant flow is provided in the Appendix A.

An analysis of missing data was conducted by comparing the participants who responded to all the assessment waves (*n* = 531) with those who had missing data for one or more assessment waves (*n* = 613). The two groups did not differ in terms of gender, *p* = 0.493. However, the participants who responded to all four assessment waves were significantly older (*M* = 45.01; *SD* = 13.97) than those who responded to two or three waves (*M* = 43.20; *SD* = 13.05); *t*(1142) = -2.27 and *p* = 0.023. According to a Mann–Whitney *U* test, the participants who responded to all waves were slightly better educated (mean rank = 597.48) than the non-respondents (mean rank = 550.87); *U* = 176,013.50, *z* = 3.08 and *p* = 0.002. Moreover, the average monthly household income was higher in respondents (mean rank = 577.13) than in non-respondents (mean rank = 539.09); *U* = 163,964.50, *z* = 2.052 and *p* = 0.040. Please note that complete cases were used for the descriptive analyses, whereas the FIML was used for the LGCMs. 

### 3.2. Cross-Country Differences and Measurement Invariance

Due to a violation of the homoscedasticity assumption, a Welch *t*-test was run to determine whether there were differences in the mean ages of the Austrian and Croatian participants. The Austrian participants were significantly older than the Croatian participants (*M* = 3.78, 95% CI [2.09, 5.47], *t*(755.11) = 4.40, *p* < 0.001). The difference in the proportions of male and female participants in the two countries was also significant (*p* = 0.007). Furthermore, the participants from Croatia reported higher levels of educational attainment (*U* = 184606.50, *z* = 8.04, *p* < 0.001), whereas the participants from Austria reported higher income (*U* = 124867.00, *z* = −3.05, *p* = 0.002).

Regarding measurement invariance, the model fit was satisfactory in the most restricted model for all measures tested (see the Appendix A), suggesting strong invariance for the ADNM-8, WHO-5, PHQ-2 and GAD-2. Thus, the results show that all the measures included in the analysis measured the same constructs over different timepoints and different groups (i.e., the Austrian and Croatian samples).

### 3.3. Mental Health Outcomes over Time

At T1, one in ten participants reported clinically relevant AD symptoms, and approximately 15% met the criteria for probable depression or an anxiety disorder. All mental health outcomes assessed in the study showed changes over time, with the highest AD and depression scores at T2 and the highest anxiety scores at baseline. At each timepoint, a considerable number of participants reported reduced well-being, with percentages ranging from 33.7% to 45.1%. Descriptive statistics of all relevant measures at each timepoint are shown in Table 3. The observed individual trajectories of the ANDM-8 and WHO-5 scores are provided in the Appendix A.

The percentages of self-reported AD differed significantly over time (χ^2^(3) = 22.71; *p* < 0.001). The highest percentage of participants at risk of developing AD was at T2 (15.4%; *n* = 82), which was significantly more than at T1 (9.2%; *n* = 49), T3 (10.4%; *n* = 55) and T4 (11.5%; *n* = 61), *p* < 0.001, *p* = 0.002 and *p* = 0.027, respectively. 

A significant difference regarding probable depression was also found (χ^2^(3) = 8.80; *p* = 0.032). The percentage of participants at risk for probable depression was equally high at T2 and T4 (16.0%; *n* = 85), followed by T1 (13.9%; *n* = 74) and T3 (11.5%; *n* = 61), though Dunn’s procedure with a Bonferroni correction could not detect where the differences in proportions lay (*p* ≥ 0.065). The percentage of participants at risk for probable anxiety changed over time (χ^2^(3) = 11.38; *p* = 0.010), with the highest prevalence rates found at T1 and T4 (14.9%; *n* = 76). These estimated prevalence rates were significantly different from those found at T3 (9.6%; *n* = 51), *p* = 0.023, but not from those found at T2 (13.4%; *n* = 71), *p* > 0.050.

Linear regression analyses showed that both depression and anxiety scores at baseline could predict AD scores at the last timepoint (*F*(1, 672) = 110.72; *p* < 0.001 and *F*(1, 672) = 122.13; *p* < 0.001, respectively). The adjusted *R*^2^ was 14.0% for depression and 15.3% for anxiety, both indicating a moderate effect size according to Cohen [70].

The depression and anxiety scores at baseline were also predictive of the WHO-5 scores at the last timepoint (*F*(1, 672) = 165.20; *p* < 0.001 and *F*(1, 672) = 150.66; *p* < 0.001, respectively). The effect size was moderate, with *R*^2^ = 19.7% for depression and *R*^2^ = 18.3% for anxiety [70]. 

### 3.4. Univariate LGCMs

The model fit indices for the univariate LGCMs showed mixed results. While the Comparative Fit Index (CFI) and Tucker–Lewis Index (TLI) mostly indicated an acceptable fit, the Root Mean Square Error of Approximation (RMSEA) and Standardised Root Mean Square Residual (SRMR) occasionally indicated a poor fit. All fit indices and parameter estimates are presented in Table 4. 

In all models, the intercept was significant, whereas the slope was significant only in the unconditional model and the model with TVCs. According to these two models, AD symptoms increased, whereas well-being scores decreased over time. The covariance between the slope and intercept did not reach statistical significance in any of the models tested. The same was true for the predictors of the slope, with only country being a nearly significant predictor of the change in well-being over time in the second model (Table 4). 

On the other hand, several intercept predictors were found. Women, participants from Austria and those with lower income showed, on average, higher baseline scores on the ADNM-8, whereas older participants, participants from Croatia and those with higher income showed higher levels of well-being at baseline. 

Regarding the time-varying predictors, both the depression and anxiety scores were predictive at all timepoints, with slightly higher estimates for anxiety in AD models and higher estimates for depression in well-being models. From the empirical–practical perspective, the partially poor fit indices and the significant variances were sufficient to justify testing additional models. 

Figure 2 shows the estimated trajectories of AD and well-being based on the unconditional LGCM. For better clarity, individual trajectories are depicted only for a randomly selected subsample; the remaining individual trajectories are provided in the Appendix A.

### 3.5. Multivariate LGCMs

The multivariate models achieved a better fit with the data than the univariate models. The intercepts for both AD and well-being reached statistical significance in the unconditional and conditional model, as presented in Table 5. The covariance between the intercepts of both constructs was significant in both models, indicating that on average, participants with higher initial AD symptoms showed lower initial levels of well-being and vice versa. The significant covariance between the slopes of both constructs suggests a negative association between the rate of change in AD symptoms and the rate of change in well-being. However, this covariance was only significant in the unconditional model. The covariances of the residuals were significant at each timepoint (*p* < 0.001). 

Several significant predictors were detected in the final conditional model with all TVCs and TICs (see Table 5 for details). Higher symptoms of AD at baseline were found in women, older participants, those coming from Austria and those with higher income, with the last one being only nearly significant. Initial well-being scores were found to be higher in older participants and those coming from Croatia. None of the slope estimates reached statistical significance. Regarding the TVCs, both anxiety and depression scores could predict AD and well-being scores at all timepoints. Again, the predictive power of the anxiety scores was higher in the case of AD, whereas the predictive power of the depression scores was higher in the case of well-being. The intercepts of both measures and the slope of AD showed significant variance in both multivariate models but were lower in the conditional model.

### 3.6. Multivariate LGCMs by Country

The proposed linear model showed no convergence issues in Austria. However, when estimating the model in the Croatian sample, a negative residual variance for the well-being slope was detected. This is not plausible, given that computation of residual variances requires squaring values which are always positive [71]. We addressed the problem by freeing two initially fixed well-being parameters as follows: 0, a, b, 1 (instead of: 0, 1, 2, 3). 

The model fit was acceptable for the Austrian and the Croatian data, as shown in Table 6. The intercept for the AD scores was slightly higher in Austria, whereas the intercept for the well-being score was higher in Croatia. The slope for the AD score was positive in the Austrian model, indicating an increase over time, while the negative slope in the Croatian model indicates a decrease over time. A decrease in the level of well-being was found in both models. The intercepts of AD and well-being were significantly negatively correlated in the Austrian model, suggesting a negative association between the ANDM-8 and WHO-5 scores at baseline. None of the tested covariances were significant in the Croatian model. 

While age and income were significant predictors of the AD intercept in the Austrian model, gender was the only intercept predictor in the Croatian model. None of the TICs were significant predictors of the change over time regarding neither AD nor well-being. 

Again, depression and anxiety were shown to be predictive of AD and well-being scores. Regarding AD, higher estimates for depression were found at T2 and T4 and higher estimates for anxiety were found at T1 and T3 in the model with Austrian participants. The opposite was found in the model with Croatian participants. Regarding well-being, the depression scores had a higher predictive power in both countries at all timepoints. 

## 4. Discussion

The present study is the first study to investigate the trajectories of AD symptoms and well-being during nearly two years of the COVID-19 pandemic. We found an association between the starting point of the two trajectories, sociodemographic characteristics and the symptoms of depression and anxiety. Furthermore, the comparison between Austrian and Croatian data allowed us to identify country-specific predictors and differences in mental health trajectories.

### 4.1. Mental Health Outcomes over Time

Approximately one in ten participants met the criteria for probable AD at T1, T3 and T4. The prevalence was higher (15.1%) at the second timepoint, which was characterised by a notable increase in the number of cases (compared to T1) and stricter government measures than at other timepoints (see Table 1 for details). The intensity of the pandemic and associated restrictions have already been linked to mental health problems [20,47] but not specifically to AD. Previous studies conducted during COVID-19 reported prevalence rates of AD ranging from 8.2% [15] and 18.8% [72] to 61.3% [14]. However, these rates are only partly comparable with our results since the data were collected in the initial stages of the pandemic (spring of 2020), whereas our study started in the summer of 2020. 

The prevalence rates of self-reported anxiety and depression ranged between 11.9% and 16.2%. This is relatively low compared to the pooled prevalence rates found in a recent umbrella review [21], which might be explained by the timing of the study. Namely, as shown in the review, prevalence rates were generally higher in studies conducted in the earlier stages of the pandemic, and the present study covered later stages of the pandemic. 

The WHO-5 scores, which point to poor well-being and the presence of depressive symptoms in more than one-third of the participants at all timepoints, are rather alarming. This might indicate subtle mental health difficulties related to COVID-19 which do not necessarily lead to a diagnosis of a mental disorder but might impact a person’s mood and challenge their ability to cope. Moreover, several studies found a link between well-being and COVID-19 [31,47].

### 4.2. Trajectories of Adjustment and Well-Being

In the first step of the analysis, univariate models were tested to estimate the trajectories of AD and well-being separately. Contrary to our expectations, three univariate models for AD (i.e., the independent trajectories of AD) showed a minor increase in symptoms over time. Only one model showed a decrease; however, this result was not significant. The current literature on adjustment during COVID-19 is scarce and limited to the initial stages of the pandemic. For instance, a German study found a minor, non-significant decrease in AD scores in the first year of the pandemic [26]. The literature on AD outside the context of COVID-19 suggests that AD symptoms can persist for longer than six months if the stressor persists longer [13]. It is thus plausible that the duration and the changing course of the pandemic, with the varying number of infections and different restrictions being imposed at different times, repeatedly challenged people’s adaptational capacities, hindering successful adjustment. 

The results regarding the trajectories of well-being support this assumption. According to the negative slopes in all the well-being models, well-being decreased over time, emphasising the difficulties in maintaining mental health over almost two years of the pandemic. Acknowledging the human ability to cope with and adapt to stressful situations [39], we hypothesised an improvement in mental health over time. However, COVID-19 posed numerous challenges in different areas of life [1], recurrently forcing people to deploy additional resources to cope and further exacerbating their mental fragility. This might explain the deterioration of mental health found in our study. A study including a post-pandemic follow-up would help to better understand adjustment in the aftermath of the COVID-19 pandemic. 

The results of the multivariate models (i.e., the joint trajectories of AD and well-being) were only partially in accordance with those of above-described univariate models. The simultaneous estimation of the trajectories of AD and well-being confirmed a decrease in well-being over time. However, the slope of AD did not reach statistical significance, suggesting a flat trajectory. This is contradictory to the rising trajectory identified in the univariate models. At this point, it must be mentioned that all the slope means in the univariate models were rather low, indicating that the estimated trajectories were only slightly steep. It is possible that the slope means in the multivariate models were not any more significant due to the increasing complexity of the models (e.g., due to the additional covariances included in the models). Future studies should also test models of different complexities but with some additional predictors to better understand the factors underlying the joint trajectories of AD and well-being. 

One noteworthy result of the multivariate models is the significant negative relationship between the initial symptoms of AD and the initial well-being scores. Thus, people with adjustment difficulties were more likely to have poor well-being at baseline, echoing the results of previous studies [43]. In addition, the slopes of well-being and AD were correlated in the unconditional model (i.e., the model without predictors), indicating that an increase in AD is linked with a decrease in well-being. However, this relationship should be treated with caution because it was not significant in the conditional model. 

Finally, the significant variances of the intercepts in all models tested suggest considerable individual variabilities in the initial levels of AD and well-being. This heterogeneity might be attributed to a combination of sociodemographic characteristics, different impacts of the pandemic at the individual level (e.g., losing a loved one or having had COVID-19) or prior mental health difficulties [73,74,75]. The fluctuations in AD symptoms also differed significantly across individuals. This is in line with a recent study which identified four distinct trajectories of AD [54], highlighting the high variability in people’s reactions to the pandemic, as demonstrated earlier [76,77]. 

### 4.3. Predictors of Adjustment and Well-Being Trajectories

In our univariate conditional models, several sociodemographic variables could significantly predict the starting points of the trajectories of AD and well-being. Higher initial AD symptoms were identified in female participants, participants from Austria and those with lower income. On the other hand, the initial well-being scores were higher in older participants, participants from Croatia and those with higher income. These results are in line with previous COVID-19 studies which reported higher risks for AD for women and people with financial difficulties [17,18]. Notably, both gender and income could predict only the starting point but not how the symptoms of AD and well-being would change over time. For this reason, it can be assumed that women and people with low income were at risk even before the pandemic, and this risk remained throughout the pandemic. This idea is supported by a recent meta-analysis which showed that female gender and low income are reliable predictors of AD in different contexts [78]. Thus, these factors cannot only be considered pandemic-specific but rather universal, pointing to the importance of providing regular, low-threshold psychosocial support for women and other vulnerable groups, such as people with low income.

In the present study, older participants tended to have higher levels of well-being but also higher AD scores at baseline. This is contrary to a Spanish study which found that the elderly adapted better to COVID-19-related restrictions but also had more worries [79]. This discrepancy could be attributed to the timing of the two studies: while the Spanish study investigated the immediate psychological impact of the first Spanish lockdown, our study started several weeks after the end of the first lockdown in Austria and Croatia. It is therefore conceivable that the role of age varies depending on the stage of the pandemic. Considering the inconsistencies in the COVID-19 literature regarding the effects of age, future studies should more closely examine age differences in adapting to new environments and possible mechanisms behind, such as emotion regulation [22]. 

The estimation of the joint trajectories of AD and well-being revealed results comparable to those of the independent trajectories. Age and country were related to the initial levels of both AD and well-being, while gender and income were related only to the initial AD score. Sociodemographic characteristics thus seemed to influence the mental health of the participants at baseline but not the trajectories of their mental health over time. 

It should be noted that education level was not a significant predictor in any of the models tested. Previous research delivered mixed results on the effect of education on AD [78]. Regarding well-being, a Danish study found a greater decline in well-being in highly educated people [31], whereas a lower level of education was associated with greater psychological distress in a German study [52]. To disentangle the relationship between education and mental health, samples in forthcoming studies should accurately represent the levels of education in the studied population.

As expected, the included time-dependent covariates, namely, the anxiety and depression scores, could significantly predict both AD and well-being scores at all timepoints in the univariate and multivariate models. While the anxiety and depression scores were positively correlated with the symptoms of AD, they were negatively correlated with well-being. These relationships are not surprising given that the symptoms of anxiety and depression were common emotional responses during past epidemics and natural disasters [80] and were repeatedly associated with AD symptoms during COVID-19 [18,23,24]. Moreover, depression and anxiety symptoms were among the most common indicators of well-being in longitudinal COVID-19 studies [30]. 

Notably, when predicting AD symptoms, higher estimates were observed for anxiety, whereas higher estimates for depression were observed when predicting well-being. According to some sources, anxiety is a normal reaction to the novel and unpredictable pandemic situation and is thus expected during the adjustment process [26,75]. Therefore, it can be assumed that AD with anxiety was the most frequent subtype of AD during COVID-19, which might explain the higher estimates for anxiety when predicting AD. On the other hand, well-being and depression are highly correlated and sometimes even perceived as opposite poles of the same continuum [45], which was possibly reflected in the stronger association between well-being and depression scores. Further research during pandemics and similar crises should assess AD subtypes and investigate the relationship between well-being and depression in a systematic way. 

### 4.4. Cross-Country Differences in Adjustment and Well-Being Trajectories 

Our study demonstrated several cross-country differences. On average, the Austrian participants had higher initial levels of AD and lower well-being. This finding is consistent with a large-scale international study which showed a greater mental health burden in Austria than in Croatia [37]. In the pandemic context, there are several possible explanations for the observed differences between the two countries. Firstly, COVID-19 severely impacted the global economy and job market [81] and, for many, achieving financial or professional goals was more difficult than prior to the pandemic. This is relevant as financial and work-related problems are commonly associated with AD [10,11]. The reduced possibility to achieve goals might have impacted the mental health of the Austrian participants more strongly given that satisfaction with life achievements was found to be the strongest predictor of well-being among Austrians [82]. Secondly, the incidence of COVID-19 was higher in Austria than in Croatia during the first assessment timepoint, which might have had an effect on the participants’ well-being and their adjustment to the pandemic at the baseline assessment. Thirdly, a large pan-European study reported that Austria was among the countries which experienced significant growth in their vulnerability levels, suggesting a greater mental health impact of COVID-19 on their population compared to some other European countries [83]. Finally, there is evidence that countries with histories of stress and trauma showed better mental health outcomes during the pandemic [19]. It can therefore be assumed that the pandemic was perceived less adversely than previous crises in Croatia (e.g., the Homeland War, 1991–1995, and major earthquakes in 2020 [84,85,86]), which might have facilitated psychological adjustment to the pandemic situation. 

When comparing the multivariate models by country, no significant change in AD and well-being over time could be found. This was not anticipated given the significant rates of change observed in previous models. This discrepancy can be attributed to the smaller sizes of the national subsamples in comparison to the overall sample which was used in the previous models. On a descriptive level, the level of well-being tended to decline over time in both countries, whereas the symptoms of AD tended to increase over time in Austria and decrease in Croatia. It is thus conceivable that not only were the Austrian participants at a higher risk for AD at baseline but they were also more likely to develop AD symptoms as the pandemic progressed. However, a note of caution is due here since the effect was not significant. Interestingly, a slightly greater variability in terms of the fluctuations in AD symptoms was observed among the Austrian participants. One plausible reason for this might lie in the stringency index [34], which varied more strongly in Austria (ranging between 38.59 and 77.58) than in Croatia (ranging between 35.19 and 58.57). It has already been suggested that the intensity of the pandemic might account for mental health differences over time [20,21]. 

In Austria, older age and lower income were related to higher initial AD symptoms, whereas female gender was related to higher initial AD symptoms in Croatia. This finding is important as it implies that the predictors of mental health amidst pandemics might be country-dependent. Another peculiar finding was that anxiety and depression had different predictive powers depending on the timepoint of assessment. Compared to depression, anxiety scores were a stronger predictor of AD at T1 and T3 in Austria and at T2 and T4 in Croatia, with the respective timepoints corresponding to higher COVID-19 incidence rates. By way of illustration, the average incidence was higher in Austria than in Croatia at T1, and at this timepoint, the estimated value for anxiety was higher than the estimated value for depression in Austria and also higher than the estimated value in Croatia. This might be due to the presence of a greater fear of infection at times of higher infection rates, resulting in higher anxiety scores. This idea is supported by several studies which demonstrated the relationship between the fear of infection and AD symptoms [18,23]. In conclusion, the observed differences between Austria and Croatia highlight the importance of national studies which would consider country-related factors, e.g., cultural values and the pandemic situation. Such studies might provide a more complete picture of the psychological adjustment to the COVID-19 pandemic, as well as to other similar crises. 

### 4.5. Strengths and Limitations

To the best of our knowledge, this is the first study to address changes in AD symptoms and well-being which was conducted in the later stages of the pandemic and in two countries. The lack of longitudinal studies is considered to be an important gap in AD research [13] which was addressed in this paper. The inclusion of both time-invariant and time-varying predictors and the examination of the co-development of AD and well-being trajectories via sophisticated methodology, was also innovative. Moreover, the present study had a longer assessment period than most COVID-19 studies, thus providing novel insights into the mental health impact of COVID-19. Another strength of our work is the investigation of AD and well-being in two countries that have rarely been jointly investigated in the past. This cross-country comparison is particularly relevant in COVID- 19 research since it adds to a better understanding of different responses to the pandemic in different contexts. Finally, a total of *N* = 531 participants responded to all four assessment waves, which is surprisingly good considering the length of the questionnaire and the long period of data collection. 

However, the generalisability of our results is subject to certain limitations. First, convenience sampling was used in both countries, and the assessment waves were conducted at slightly different times (see Table 1). Second, the Austrian and Croatian samples were not of the same size, and they differed in sociodemographic characteristics, which might have biased the results. Third, the model fit was only satisfactory for some of the models tested, with the best model fit being found for the unconditional multivariate model. Future studies should include different covariates to assess their predictive power and check whether this could improve the model fit. Fourth, the Heywood case, one common problem of LGCMs, occurred when testing the final model in the Croatian sample. We managed to resolve the problem by freeing some of the previously fixed parameters in the Croatian model only, which might have limited its comparability with the Austrian model. Furthermore, pre-pandemic data and the data collected during the very first stage of the pandemic would have provided additional insights into the development of AD symptoms, but these data were not available for the countries investigated. Since the exploration of different subgroups using a Latent Class Analysis or Latent Profile Analysis was beyond the scope of our study, this might be an interesting topic for future research. Lastly, timestamps for individual survey responses were not available in our surveys, which is why it was not possible to consider the values of the stringency index and the incidence and death rates for all the dates during the data assessment period. This approach could be used in forthcoming investigations.

## 5. Conclusions

The present study sought to examine the trajectories of AD and well-being across Austria and Croatia and the factors underlying these trajectories. By focusing on mental health and well-being, our study makes a modest but meaningful contribution to achieving the third goal of the 2030 Agenda for Sustainable Development [87]. We found gender, age and income, as well as anxiety and depression scores, to be predictive of AD and the level of well-being during the COVID-19 pandemic, with the predictive powers of these covariates varying in the two countries. According to our results, mental health did not improve over time, highlighting the importance of psychosocial support during pandemics and similar global crises. In times of social restrictions, therapist-assisted computerised interventions might be particularly helpful [88,89]. Importantly, our results reveal a great variability in people’s responses to the pandemic, supporting the context-sensitive perspective in mental health research and clinical practice. Special attention should be paid to vulnerable groups, such as women and people with financial difficulties, to avoid further loss spirals and the development of full-blown mental disorders.

## Figures and Tables

**Figure 1 ijerph-20-06861-f001:**
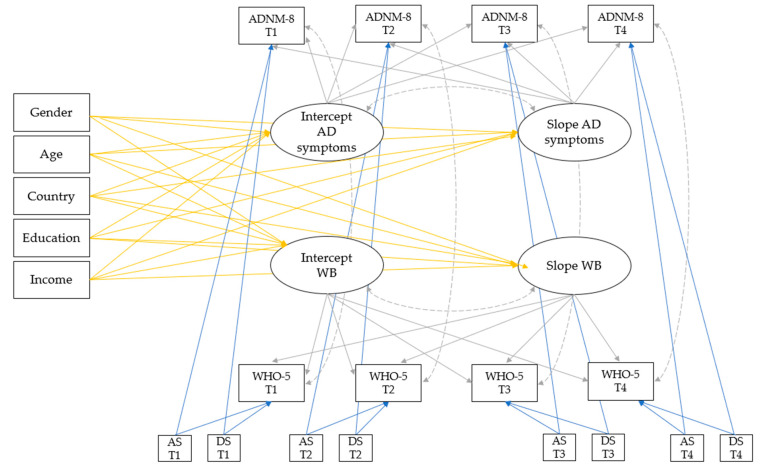
Conceptional model of AD symptoms and well-being with covariances, time-invariant covariates (TICs; yellow lines) and time-varying covariates (TVCs; blue lines). The solid grey lines represent the factor loadings, and the dashed grey lines represent the covariances. Note: ADNM-8 = 8-item Adjustment Disorder New Module; AD = adjustment disorder; WB = well-being; AS = anxiety scores (PHQ-2 = 2-item Patient Health Questionnaire); DS = depression score (GAD-2 = 2-item Generalized Anxiety Disorder Scale).

**Figure 2 ijerph-20-06861-f002:**
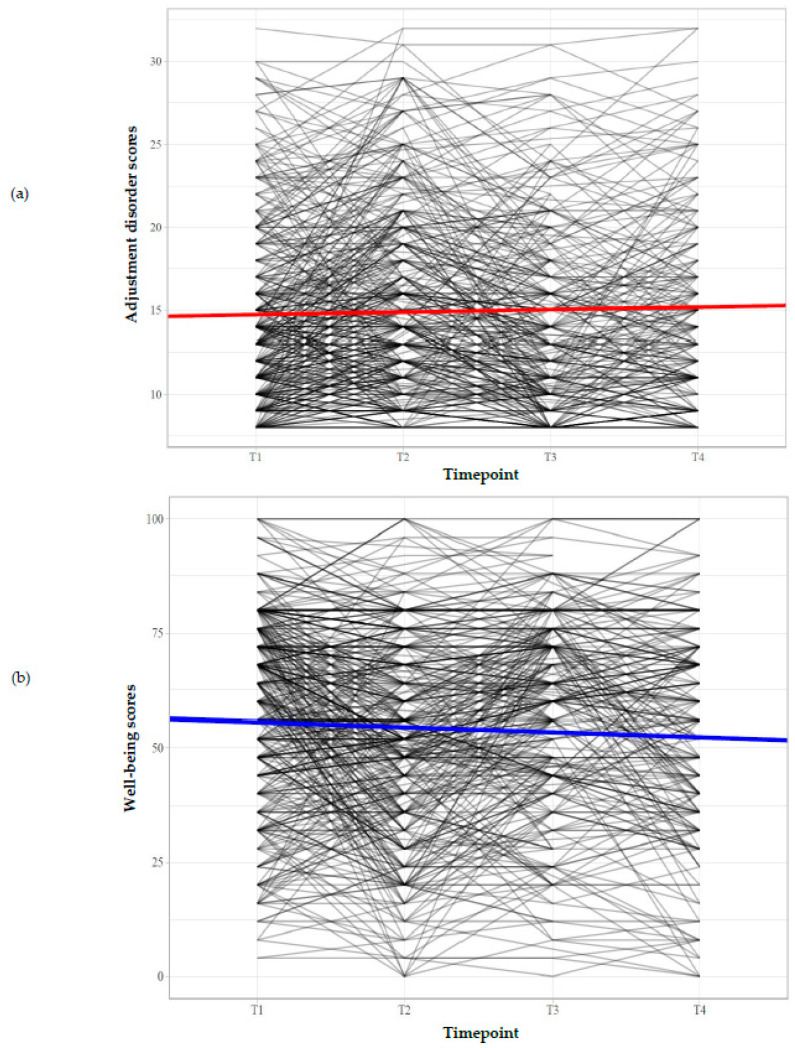
The estimated growth trajectories based on the unconditional univariate models. (**a**) Adjustment disorder trajectory. (**b**) Well-being trajectory.

**Table 1 ijerph-20-06861-t001:** The COVID-19 situation in Austria and Croatia at four data collection timepoints.

	T1		T2		T3		T4	
	Austria	Croatia	Austria	Croatia	Austria	Croatia	Austria	Croatia
Recruitment period	27 June 2020– 22 September 2020	15 June 2020– 16 August 2020	14 January 2021– 29 March 2021	30 November 2020– 7 January 2021	13 July 2021– 8 October 2021	21 June 2021– 26 July 2021	26 November 2021– 13 December 2021 ^a^	8 December 2021– 11 January 2022
Duration of data collection	88 days ≈ 13 weeks	63 days ≈ 9 weeks	75 days ≈ 11 weeks	39 days ≈ 6 weeks	88 days ≈ 13 weeks	36 days ≈ 5 weeks	18 days ≈ 3 weeks	35 days ≈ 5 weeks
Stringency index								
*M* (*SD*)	38.59	44.98	77.58	58.57	50.11	35.19	51.70	36.97
Range	36.11–50.00	35.19–54.63	73.15–82.41	47.22–67.59	46.23–55.05	30.49–41.67	49.30–52.06	33.26–38.08
Incidence								
*M* (*SD*)	24.73 (20.42)	14.55 (7.20)	214.75 (67.75)	639.83 (243.16)	126.58 (72.32)	22.81 (5.40)	413.46 (418.22)	613.41 (350.14)
Range	4.25–83.34	0.18–31.09	142.82–351.49	276.26–905.02	17.45–232.98	16.91–36.37	0.00–1257.42	0.00–981.76
Deaths								
*M* (*SD*)	0.19 (0.06)	0.22 (0.19)	4.57 (1.38)	16.35 (1.92)	0.59 (0.43)	0.59 (0.29)	7.50 (0.34)	12.15 (1.88)
Range	0.06–0.37	0.00–0.60	3.02–7.34	12.73–19.57	0.16–1.37	0.25–1.49	6.76–7.86	9.39–15.24

Note: The stringency index is a composite measure of government policies, with higher scores indicating a stricter response (i.e., 100 = strictest response [9]). Data on incidence are defined as new confirmed cases of COVID-19 per one million people. Data on death rates are defined as new deaths attributed to COVID-19 per one million people. The values reported in the table are the averages of the daily values recorded during the respective recruitment periods. All data are based on the dataset provided by *Our World in Data* (last accessed on 23 March 2023 [34]). ^a^ This data collection period was shorter than the others in order to reflect the duration of the lockdown implemented in Austria in the winter of 2021.

**Table 2 ijerph-20-06861-t002:** Participants’ characteristics at T1.

	Total Sample *N* = 1144	Austria *n* = 415	Croatia *n* = 729
Age	*M* = 44.0 *SD* = 13.5	*M* = 46.5 *SD* = 14.7	*M* = 42.7 *SD* = 12.6
	*n* (%)	*n* (%)	*n* (%)
Gender ^a^			
Male	300 (26.2)	128 (30.8)	172 (23.6)
Female	841 (73.6)	285 (68.7)	556 (76.4)
Other	2 (0.2)	2 (0.5)	0 (0.0)
Education			
Low	6 (0.5)	5 (1.2)	1 (0.1)
Middle	306 (26.7)	166 (40.0)	140 (19.2)
High	832 (72.7)	244 (58.8)	588 (80.7)
Income ^b^			
Very low	126 (11.3)	43 (11.2)	83 (11.4)
Low	346 (31.1)	140 (36.5)	206 (28.3)
Medium	344 (30.9)	31 (8.1)	313 (43.0)
High	296 (26.6)	170 (44.3)	126 (17.3)
Relationship status			
Single	269 (23.5)	102 (24.6)	167 (22.9)
In a relationship	875 (76.5)	313 (75.4)	562 (77.1)
Employment status ^c^			
Training/Study	124 (25.5)	52 (12.5)	72 (9.9)
Employed part-time	154 (34.5)	122 (29.4)	32 (4.4)
Employed full-time	742 (77.0)	193 (46.5)	549 (75.3)
Self-employed	68 (15.2)	36 (8.7)	32 (4.4)
Retired	84 (19.0)	57 (13.7)	27 (3.7)
Job-seeking	57 (12.3)	8 (1.9)	49 (6.7)
Other	26 (6.0)	10 (2.4)	16 (2.2)
Diagnosis of a mental disorder			
Yes	167 (14.6)	96 (23.1)	71 (9.7)
No	977 (85.4)	319 (76.9)	658 (90.3)

Note: ^a^ *N* = 1143 (*n_Austria_* = 415; *n_Croatia_* = 728). ^b^ *N* = 1112 (*n_Austria_* = 384; *n_Croatia_* = 728). ^c^ Percentages sum to more than 100 as multiple responses were possible.

**Table 3 ijerph-20-06861-t003:** Average scores of the assessed mental health outcomes and self-reported prevalence rates (i.e., the percentage of participants above the cut-off) of mental health disorders at four timepoints.

Timepoint	ADNM-8		PHQ-2		GAD-2		WHO-5 ^a^	
	*M* (*SD*)	% (*n*)	*M* (*SD*)	% (*n*)	*M* (*SD*)	% (*n*)	*M* (*SD*)	% (*n*)
**T1** *n* = 1144	14.3 (5.3)	9.8 (112)	1.3 (1.5)	15.4 (176)	1.4 (1.3)	15.8 (181)	58.0 (21.4)	33.7 (386)
**T2** *n* = 993	15.8 (5.7)	15.1 (150)	1.5 (1.5)	16.2 (161)	1.3 (1.5)	15.5 (154)	51.5 (22.1)	45.1 (448)
**T3** *n* = 837	14.3 (5.4)	9.9 (83)	1.2 (1.4)	12.9 (108)	1.1 (1.4)	11.6 (97)	57.3 (21.6)	34.5 (289)
T4 *n* = 674	15.0 (5.6)	11.7 (79)	1.3 (1.6)	16.2 (109)	1.3 (1.5)	14.5 (98)	53.8 (22.7)	40.8 (275)

Note: ADNM-8 = 8-item Adjustment Disorder New Module; WHO-5 = 5-item World Health Organization Well-Being Index; PHQ-2 = 2-item Patient Health Questionnaire; GAD-2 = 2-item Generalized Anxiety Disorder Scale. The prevalence rates were calculated based on the following established cut-off scores: ADNM-8 > 22, PHQ-2 > 2. GAD-2 > 2 and WHO-5 ≤ 50. ^a^ The percentages suggest suboptimal well-being and likely depression [46].

**Table 4 ijerph-20-06861-t004:** Comparison of fit indices and parameter estimates in univariate models.

		Univariate Unconditional Model		Univariate Conditional Model with TICs			Univariate Conditional Model with TVCs		Univariate Model with TICs and TVCs	
		AD	WB	AD		WB	AD	WB	AD	WB
**Fit indices**										
Χ^2^ (*df*)		105.89 *** (5)	136.80 *** (5)	131.84 *** (15)		158.36 *** (15)	148.70 *** (29)	140.68 *** (29)	169.82 *** (39)	147.65 *** (39)
CFI		0.933	0.925	0.929		0.924	0.925	0.940	0.919	0.941
TLI		0.919	0.910	0.876		0.869	0.901	0.921	0.879	0.912
RMSEA [90% CI]		0.165 [0.136, 0.196]	0.182 [0.153, 0.213]	0.101 [0.084, 0.119]		0.107 [0.091, 0.125]	0.097 [0.082, 0.113]	0.091 [0.076, 0.107]	0.089 [0.075, 0.103]	0.078 [0.065, 0.092]
SRMR		0.063	0.068	0.034		0.037	0.119	0.106	0.085	0.070
**Parameter estimates** (*SE*)										
Intercept		14.57 *** (0.16)	56.77 *** (0.64)	16.34 *** (1.28)		39.88 *** (4.70)	11.73 *** (0.22)	69.32 *** (0.85)	10.78 *** (1.34)	61.49 *** (5.17)
Slope		0.15 * (0.06)	−1.10 *** (0.22)	0.17 (0.47)		−2.76 (1.67)	0.24 ** (0.09)	−1.34 *** (0.30)	−0.33 (0.56)	−0.76 (1.84)
**Covariances** (*SE*)										
i ~~ s		−0.78 (0.51)	8.31 (6.71)	−0.97 ^†^ (0.50)		5.98 (6.86)	0.17 (0.37)	4.04 (5.05)	0.07 (0.37)	2.85 (5.10)
**Variances** (*SE*)										
Intercept		18.87 *** (1.38)	286.64 *** (19.74)	18.31 *** (1.36)		274.99 *** (20.02)	7.13 *** (1.11)	114.14 *** (14.58)	6.52 *** (1.16)	115.11 *** (14.53)
Slope		1.24 *** (0.30)	3.66 (3.80)	1.30 *** (0.29)		3.78 (3.82)	0.59* (0.23)	0.83 (2.88)	0.58 * (0.23)	1.47 (2.94)
**TICs** (*SE*)										
Gender		-	-	i: s:	1.25 ** (0.37) 0.04 (0.13)	−1.55 (1.48) −0.52 (0.49)	-	-	1.10 ** (0.39) 0.04 (0.15)	−0.88 (1.60) −0.09 (0.51)
Age		-	-	i: s:	−0.00 (0.01) 0.01 (0.00)	0.22 *** (0.05) −0.00 (0.02)	-	-	0.04 ** (0.01) 0.01 (0.01)	0.11 * (0.05) −0.01 (0.02)
Country		-	-	i: s:	−1.12 ** (0.35) −0.09 (0.14)	4.04 ** (1.36) 0.93 ^†^ (0.47)	-	-	−0.81 * (0.38) −0.00 (0.15)	2.75 ^†^ (1.40) −0.42 (0.50)
Education		-	-	i: s:	−0.64 (0.39) −0.08 (0.16)	−0.01 (1.52) 0.57 (0.56)	-	-	−0.42 (0.44) 0.05 (0.18)	0.51 (1.76) 0.34 (0.58)
Income		-	-	i: s:	−0.51 ** (0.17) −0.01 (0.07)	1.86 ** (0.67) 0.08 (0.23)	-	-	−0.36 ^†^ (0.19) 0.03 (0.08)	−0.57 (0.70) 0.14 (0.25)
**TVCs** (*SE*)										
T1	Depression Anxiety	-	-	-		-	1.00 *** (0.17) 1.01 *** (0.17)	−5.28 *** (0.52) −3.25 *** (0.48)	0.98 *** (0.17) 1.00 *** (0.18)	−4.99 *** (0.53) −3.42 *** (0.48)
T2	Depression Anxiety	-	-	-		-	1.29 *** (0.17) 1.31 *** (0.18)	−6.31 *** (0.54) −4.54 *** (0.53)	1.26 *** (0.18) 1.32 *** (0.18)	−6.30 *** (0.55) −4.46 *** (0.55)
T3	Depression Anxiety	-	-	-		-	0.85 *** (0.17) 1.27 *** (0.17)	−5.63 *** (0.62) −2.97 *** (0.67)	0.94 *** (0.17) 1.16 *** (0.17)	−5.39 *** (0.64) −3.23 *** (0.68)
T4	Depression Anxiety	-	-	-		-	0.94 *** (0.14) 1.01 *** (0.15)	−6.03 *** (0.58) −3.23 *** (0.59)	0.96 *** (0.15) 1.04 *** (0.16)	−5.92 *** (0.62) −3.36 *** (0.62)

Note: TICs = time-invariant covariates; TVCs = time-varying covariates; AD = adjustment disorder; WB = well-being; CFI = Comparative Fit Index; TLI = Tucker–Lewis Index; RMSEA = Root Mean Square Error of Approximation; CI = Confidence Interval; SRMR = Standardised Root Mean Square Residual; SE = Standard Error; i = intercept. s = slope. The robust CFI, TLI and RMSEA, along with the scaled SRMR, are reported. The literature suggests considering a lower CI limit CI when interpreting the RMSEA [60,65]; conventionally, values of 0.05–0.08 indicate an acceptable fit, and values ≤ 0.05 indicate a very good fit [58]. ^†^
*p* = 0.05. * *p* < 0.05. ** *p* < 0.01. *** *p* < 0.001.

**Table 5 ijerph-20-06861-t005:** Comparison of fit indices and parameter estimates in the multivariate models.

		Multivariate Unconditional Model ^a^		Multivariate Conditional Model (All Predictors)	
**Fit indices**					
Χ^2^ (*df*)		188.49 *** (21)		310.69 *** (89)	
CFI		0.978		0.937	
TLI		0.971		0.906	
RMSEA [90% CI]		0.081 [0.071, 0.092]		0.075 [0.066, 0.084]	
SRMR		0.060		0.103	
**Parameter estimates** (*SE*)					
Intercept		adnm_i: 14.58 *** (0.22)	well_i: 57.33 *** (0.86)	adnm_i: 10.86 *** (1.36)	well_i: 61.39 *** (5.21)
Slope		adnm_s: 0.10 (0.07)	well_s: −1.03 *** (0.25)	adnm_s: −0.35 (0.56)	well_s: −0.92 (1.83)
**Covariances** (*SE*)					
adnm_i ~~ adnm_s		−0.85 (0.55)		0.09 (0.37)	
well_i ~~ well_s		9.08 (6.69)		3.02 (5.04)	
adnm_i ~~ well_i		−52.34 *** (5.45)		−7.41 * (3.10)	
adnm_s ~~ well_s		−1.19 * (0.55)		0.04 (0.38)	
**Variances** (*SE*)					
Intercept		adnm: 20.18 *** (1.94)	well: 277.56 *** (24.23)	adnm: 6.65 *** (1.18)	well: 116.78 *** (14.73)
Slope		adnm: 1.33 *** (0.35)	well: 4.66 (3.99)	adnm: 0.52 * (0.23)	well: 0.98 (2.88)
**Time-invariant covariates** (*SE*)					
Gender		-	-	adnm_i: 1.13 ** (0.40) adnm_s: 0.03 (0.15)	well_i: −0.94 (1.60) well_s: −0.07 (0.52)
Age		-	-	adnm_i: 0.04 ** (0.01) adnm_s: 0.01 (0.01)	well_i: 0.11 * (0.05) well_s: −0.01 (0.02)
Country		-	-	adnm_i: −0.82 * (0.38) adnm_s: 0.01 (0.15)	well_i: 2.74 ^†^ (1.40) well_s: −0.39 (0.50)
Education		-	-	adnm_i: −0.43 (0.44) adnm_s: 0.05 (0.18)	well_i: 0.54 (1.76) well_s: 0.32 (0.58)
Income		-	-	adnm_i: −0.37 ^†^ (0.19) adnm_s: 0.03 (0.08)	well_i: −0.56 (0.71) well_s: 0.16 (0.25)
**Time-varying covariates** (*SE*)				Adjustment disorder	Well-being
T1	Depression Anxiety			0.96 *** (0.18) 0.98 *** (0.18)	−4.95 *** (0.53) −3.37 *** (0.49)
T2	Depression Anxiety			1.24 *** (0.18) 1.30 *** (0.18)	−6.26 *** (0.56) −4.38 *** (0.56)
T3	Depression Anxiety			0.93 *** (0.17) 1.15 *** (0.17)	−5.31 *** (0.63) −3.15 *** (0.68)
T4	Depression Anxiety			0.94 *** (0.15) 1.04 *** (0.16)	−5.86 *** (0.61) −3.22 *** (0.61)

Note: CFI = Comparative Fit Index; TLI = Tucker–Lewis Index; RMSEA = Root Mean Square Error of Approximation; CI = Confidence Interval; SRMR = Standardised Root Mean Square Residual; SE = Standard Error; i = intercept; s = slope. The robust CFI, TLI and RMSEA, along with the scaled SRMR, are reported. The literature suggests considering a lower CI limit CI when interpreting the RMSEA [60,65]; conventionally, values of 0.05–0.08 indicate an acceptable fit, and values ≤ 0.05 indicate a very good fit [58]. ^a^ Since some estimated variances were negative in the model using the MLR as an estimator, the WLSMV was used as an alternative estimator. ^†^
*p* = 0.05. * *p* < 0.05. ** *p* < 0.01. *** *p* < 0.001.

**Table 6 ijerph-20-06861-t006:** Comparison of multivariate models by country.

		Austria		Croatia ^a^	
**Fit indices**					
Χ^2^ (*df*)		185.19 *** (85)		229.42 *** (84)	
CFI		0.932		0.938	
TLI		0.901		0.908	
RMSEA [90% CI]		0.081 [0.064, 0.097]		0.076 [0.064, 0.088]	
SRMR		0.109		0.112	
**Parameter** **estimates** (*SE*)					
Intercept		adnm_i: 9.93 *** (2.00)	well_i: 61.37 *** (7.59)	adnm_i: 9.44 *** (1.50)	well_i: 76.33 *** (6.85)
Slope		adnm_s: 0.12 (0.91)	well_s: −3.28 (2.49)	adnm_s: −0.84 (0.57)	well_s: −7.05 (4.81)
**Covariances** (*SE*)					
adnm_i ~~ adnm_s		−0.13 (0.71)		0.10 (0.39)	
well_i ~~ well_s		−2.82 (10.05)		−3.90 (29.67)	
adnm_i ~~ well_i		−13.05 * (6.59)		−2.72 (3.10)	
adnm_s ~~ well_s		0.40 (0.77)		−0.72 (1.53)	
**Variances** (*SE*)					
Intercept		adnm: 6.93 ** (2.17)	well: 156.40 *** (28.84)	adnm: 6.46 *** (1.27)	well: 100.84 ** (34.12)
Slope		adnm: 0.85 ^†^ (0.45)	well: 3.37 (5.56)	adnm: 0.36 (0.21)	well: 16.97 (34.22)
**Time-invariant** **covariates** (*SE*)					
Gender		adnm_i: 0.98 (0.66) adnm_s: −0.14 (0.27)	well_i: 2.27 (2.49) well_s: 1.25 (0.83)	adnm_i: 1.09 * (0.48) adnm_s: 0.23 (0.16)	well_i: −4.37 (2.38) well_s: −1.08 (1.87)
Age		adnm_i: 0.06 ** (0.02) adnm_s: 0.00 (0.01)	well_i: 0.08 (0.08) well_s: −0.02 (0.02)	adnm_i: 0.03 (0.02) adnm_s: 0.01 (0.01)	well_i: 0.07 (0.07) well_s: 0.04 (0.05)
Education		adnm_i: −0.62 (0.63) adnm_s: −0.01 (0.30)	well_i: 0.82 (2.43) well_s: 0.13 (0.80)	adnm_i: −0.35 (0.62) adnm_s: 0.14 (0.21)	well_i: 0.90 (2.62) well_s: −0.12 (1.92)
Income		adnm_i: −0.64 * (0.28) adnm_s: 0.08 (0.12)	well_i: −0.14 (1.02) well_s: 0.07 (0.33)	adnm_i: −0.08 (0.25) adnm_s: −0.05 (0.09)	well_i: −1.31 (1.06) well_s: 1.25 (0.91)
**Time-varying** **covariates** (*SE*)		Adjustment disorder	Well-being	Adjustment disorder	Well-being
T1	Depression Anxiety	0.90 ** (0.31) 1.54 *** (0.31)	−5.36 *** (0.83) −5.09 *** (0.85)	0.93 *** (0.21) 0.75 *** (0.19)	−5.00 *** (0.68) −3.42 *** (0.61)
T2	Depression Anxiety	1.61 *** (0.26) 1.18 *** (0.26)	−7.72 *** (0.85) −4.33 *** (0.84)	0.92 *** (0.23) 1.45 *** (0.25)	−4.24 *** (0.70) −3.55 *** (0.74)
T3	Depression Anxiety	0.75 * (0.29) 1.70 *** (0.25)	−5.21 *** (0.82) −4.57 *** (1.01)	1.10 *** (0.20) 0.69 ** (0.21)	−5.55 *** (0.88) −2.07 ** (0.94)
T4	Depression Anxiety	1.37 *** (0.26) 1.11 *** (0.27)	−6.59 *** (1.02) −3.27 *** (0.94)	0.53 ** (0.19) 1.03 *** (0.20)	−5.24 *** (0.73) −3.59 *** (0.78)

Note: CFI = Comparative Fit Index; TLI = Tucker–Lewis Index; RMSEA = Root Mean Square Error of Approximation; CI = Confidence Interval; SRMR = Standardised Root Mean Square Residual; SE = Standard Error; i = intercept; s = slope. The robust CFI, TLI and RMSEA, along with the scaled SRMR, are reported. The literature suggests considering a lower CI limit CI when interpreting the RMSEA [60,65]; conventionally, values of 0.05–0.08 indicate an acceptable fit, and values ≤ 0.05 indicate a very good fit [58]. Since some estimated variances were negative in the model utilizing the MLR as an estimator, the WLSMV was used as an alternative estimator. ^a^ A free-loading model was tested, with the following loadings for the slope: 0, a, b, 1. ^†^
*p* = 0.06. * *p* < 0.05. ** *p* < 0.01. *** *p* < 0.001.

## Data Availability

The data presented in this study and the R codes for the main analysis are available upon request from the corresponding author. The data are not publicly available since an additional assessment wave is planned. As long as data collection and analysis are in process, excerpts of the data can only be shared upon reasonable request and if all authors agree.

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
