# Peer review of "Trajectories of Adjustment Disorder and Well-Being in Austria and Croatia during 20 Months of the COVID-19 Pandemic"

_ijerph, 2023, doi:10.3390/ijerph20196861_

Round 1

Reviewer 1 Report

The current study aimed at investigating trajectories of adjustment disorder (AD)  and well-being over 20 months of the COVID-19 pandemic, in order to identify the  predictors of these trajectories and possible differences between Austria and Croatia.

The paper provides scientific contribution to the wide panorama of studies investigating on mental health of general population during the Covid-19 Pandemic.

The topic is interesting enough and qualitatively good to be published. This study comprehensively analysed trajectories in a very large sample. However, some changes and explanations to improve quality of manuscript for his acceptance should be done. The clarity of the manuscript is good enough in the language style, syntax, and sentence construction.

Abstract section

In the ‘abstract section’ the authors should better describe and define the objectives which appear to be three (1)….2)…3)…). Furthermore, it is not clear how the sample from general population was recruited.

Introduction section

In the ‘introduction section’ the bibliography is appropriate and updated. The authors could clarify the relationship between the Covid-19 Pandemic and trauma.

According to recent literature, the Covid-19 Pandemic represents a new type of trauma (reference listed below) due to the following characteristics: uncontrolled invisibility, persistence of traumatic distress, prevailing temporal perspective focused on the present and the future, complexity due to various components, such as fear of death, economic difficulties, lockdown and isolation, the significant changes in everyday family, social and work life.

Regarding the latter, for example a recent Italian study through qualitative and quantitative data aimed to identify potential predictors of traumatic distress during Covid -19 by focusing on emotional and cognitive correlates of social confinement in a sample di young adults (listed below).

Methods

In the ‘Methods section’, specifically ‘Procedure and Participants’ section “ SurveyLime  tool” is mentioned, without a clear description and explanation. This study is part of a large Project

involving 11 European countries. Could the authors explain why the interest was focused only on Austria and Croatia?

The authors are asked to provide clarification to the readers. What do the authors mean by "among staff of major companies and professional associations"? Are they perhaps scientific societies? Companies in which sectors? Greater clarity is also required in relation to this.

The authors should better clarify the information requested in the sociodemographic section of the administered battery. Is it just about gender, education and age? Employment status and marital status are not included. The authors, as part of this large study project, report data from a large sample analyzed from the general population. It would have been useful to know the presence of pre-existing vulnerability factors  as previous traumatic events or histories of mental illness. The authors are asked to explain the lack of interest in these variables.

In the discussion it would be useful to report potential effective interventions aimed at supporting people experiencing social traumatic  experiences, also with the support of digital technologies (references listed below).

In the ‘conclusion section’ the authors should report how their study contributes to achieving the Global Goals of the 2030 Agenda for Sustainable Development.

Listed references

https://pubmed.ncbi.nlm.nih.gov/34248442/

https://www.ncbi.nlm.nih.gov/pmc/articles/PMC7770221/

https://pubmed.ncbi.nlm.nih.gov/32055452/

https://www.thelancet.com/journals/lanpsy/article/PIIS2215-0366(23)00181-5/fulltext

Reviewer 2 Report

This study is an important contribution to the body of knowledge on COVID19, well-being and mental disorders. In my opinion, the authors have done a novel study on well-being and AD. I really enjoyed reading the paper and look forward to citing it in my own work. 

I have a few comments which I hope will improve the paper:

In the introduction the authors write about AD being a typical response and I would like to suggest an expansion on that typicalness for better coherence & meaning. I would also suggest that the authors bring in the def of AD earlier on in the introduction to anchor the section before then expanding on AD amidst COVID19 - again this will improve flow & better grounding for the reader. 

On page 2, line 47, the authors write about "by way of illustration" and I was thinking that what follows is issues around context of Croatia and Austria rather than illustration. So, essentially, I'm suggesting changing the word illustration to context. 

Initially I had a comment on the research questions (RQs) reframed as objectives, but I think the authors should leave the RQs as they are. 

I do think there needs to be better clarity on the secondary analysis component of the study because some sections in the methods are written as though this study is the primary one. So for example, Procedures and Participants should begin with, In the primary study all assessment...I base this comment on what the authors indicate on page 3, line 145, The data used for the secondary analysis ...were collected...It is a little confusing in the way it is presented currently.

I did not see a reference to Figure 1 (Conceptual model of AD symptoms and well-being with covariances...) in the preceding text.

Overall, this was a well-written elegantly presented paper on trajectories of well-being and AD. The discussion aligns with the research questions of this study. Moreover, the authors' interpretations of the findings are logical and well-supported by the data on well-being and AD. The authors have adequately outlined the strengths and limitations of the study and what - in particular the limitations - means for the generalizability of the study. They have also emphasised some of the issues with the model fit and issues with LGCMs (Heywood case).

One typo I picked up on (hence my review of minor editing required) is on page 16, section 4.3. trajectoris = trajectories
